# Delayed Onset Bilateral Papilledema in a Young Boy’s Eyes after Trauma

**DOI:** 10.3390/medicina58010140

**Published:** 2022-01-17

**Authors:** Ting-Yi Lin, Ke-Hung Chien

**Affiliations:** Department of Ophthalmology, Tri-Service General Hospital, National Defense Medical Center, Taipei 11490, Taiwan; pa735210@gmail.com

**Keywords:** head trauma, cerebral venous sinus thrombosis, papilledema, increased intracranial pressure, dyschromatopsia

## Abstract

Cerebral venous sinus thrombosis (CVST) is a rare venous thromboembolic disease that affects young adults in their thirties, with a female predilection. Head trauma accounts for only 1–3% of cases among possible etiologies. Here, we present a particular case of trauma-related CVST with delayed-onset symptoms and signs in a young boy. A 12-year-old boy presented to the emergency department with non-specific visual symptoms 11 days after head trauma. Apart from mild-grade disc swelling in the right eye and dyschromatopsia in both eyes, no significant findings were revealed during physical examinations and a non-contrast cranial computed tomography (CT) scan. Unfortunately, the patient suffered multiple seizure attacks the following day. Trauma-related CVST, complicated by delayed-onset increased intracranial pressure, and bilateral papilledema were finally diagnosed. Physicians need increased awareness of a possible CVST diagnosis if a patient with a history of head trauma shows persistent or worsening neurological symptoms despite negative results on serial non-contrast cranial CT scans.

## 1. Introduction

Cerebral venous sinus thrombosis (CVST) is a rare venous thromboembolic disease that typically affects young adults in their third decade, with a female predilection. The male-to-female ratio of CVST is 1:1.7 in the Asian population [1]. It is a potentially life-threatening disease with an overall mortality rate of approximately 10% [2].

Many risk factors have been associated with CVST, such as pregnancy and the puerperium, oral contraceptives, inherited or acquired hypercoagulable disorders, vasculitis, surgery, head trauma, malignancy, and infection [3]. The international Study on Cerebral Vein and Dural Sinus Thrombosis reported at least one identifiable risk factor in approximately 88% of the included patients and multiple (>1) risk factors in approximately 44% [4]. In this study, we report the case of a young boy who initially presented with non-specific visual symptoms and was finally diagnosed with trauma-related delayed-onset bilateral papilledema and CVST.

## 2. Case Presentation

A 12-year-old Taiwanese boy with a history of bilateral myopia presented to the emergency department with a 1 day history of blurred vision in the right eye and color vision deficiency revealed at a local ophthalmic clinic. Eleven days before presentation, he had a forehead contusion from slipping and hitting a plastic door at school. Headache, mild nausea, and vomiting developed in the following week. He then underwent non-contrast computed tomography (CT) of the head and physical examinations by a neurologist without positive findings 9 days after the head trauma. The symptoms were mildly relieved after treatment with analgesics and antiemetics.

On the external examination, the best-corrected visual acuity (BCVA) revealed 6/6 in both eyes. The outer appearance, intraocular pressure, and biomicroscopic slit-lamp examination revealed non-significant findings. However, binocular diplopia with fluctuation was reported by the patient. The cover–uncover test showed bilateral small-angle exophoria with fluctuation. There were no limitations on eye movement in any direction of gaze, and normal pupil light reflex without relative afferent pupillary defect (RAPD) in either eye. The Ishihara color plate test revealed three wrongs in the right eye, and four wrongs in the left eye out of 15 numberplates. Dilated fundus examination showed hyperemia of the disc with mild disc swelling in the right eye and negative findings in the left eye (Figure 1a). A second non-contrast CT scan of the brain and orbital region was performed without apparent abnormalities in the emergency department. As a result, further ophthalmic follow-up was arranged 2 days later.

Unfortunately, the patient suffered multiple seizure attacks, left gaze deviation, and left lower limb numbness the following day. A cranial CT scan with contrast showed venous thrombosis in the superior sagittal sinus and frontal veins on the venography. CVST complicated with delayed onset increased intracranial pressure (IICP), papilledema, and focal seizure was diagnosed. The patient was then admitted to the pediatric intensive care unit. No remarkable findings were noted in the patient’s personal and family history. Laboratory examination of protein C, protein S, lupus anticoagulant testing by diluted Russell’s viper venom test, and peripheral blood smear revealed negative findings. The antithrombin activity level was 53.1% (the reference range in our hospital was 75–125%). After admission, an antiepileptic agent with levetiracetam (500 mg every eight hours intravenously), an antithrombotic agent with enoxaparin (total daily dose of 13,200 IU subcutaneously), and mannitol (70 g every six hours intravenously) were prescribed. Bilateral papilledema (Frisen grade 2 in the right eye and grade 1 in the left eye) was revealed on the follow-up dilated fundus examination 2 days after. Brain MRI with contrast further confirmed the diagnosis of venous sinus thrombosis in the superior sagittal sinus (Figure 2).

The patient’s symptoms improved substantially after the treatment. No further dyschromatopsia was found in either eye. However, mild visual field (VF) deficits, especially over the left quadrant in both eyes, were revealed on the Humphrey central 30-2 threshold test (Figure 1b). After treatment for 1 month, the patient was discharged with persistent bilateral papilledema and intact visual acuity (BCVA 6/6, bilaterally). He was prescribed oral levetiracetam 500 mg every 12 h and enoxaparin 15,000 IU daily, both of which were gradually tapered during outpatient visits. At the 3 month follow-up, his condition remained stable with partial resolution of the bilateral papilledema (Frisen grade 1, bilaterally; Figure 1c). The papilledema was almost completely resolved 10 months after disease onset without sequelae (mild Frisen grade 1 in the right eye; complete resolution in the left eye; Figure 1d).

## 3. Discussion

The clinical presentation of CVST is highly variable among patients and is not specific. Headache (88.8%), seizures (39.3%), paresis (37.2%), papilledema (28.3%), and mental status changes (22%) are some of the most commonly presented symptoms [4]. Treatment of CSVT generally involves supportive care, symptom-relief measures, anticonvulsants to control seizures, antibiotics for underlying infection, antithrombotic agents, and medical or surgical measures for decreasing IICP [5]. Ocular manifestations of IICP typically include bilateral disc edema, transient visual obscuration, enlarged blind spot on VF testing, minimal to no RAPD, minimal to no dyschromatopsia, obliterated cupping, and absent of spontaneous venous pulse [6].

The average time to resolution of papilledema in patients with CVST is approximately 6 months. Even though most patients have good final visual acuity, a significant proportion (40%) suffer permanent visual field deficits. The risk of visual field loss increases with more severe papilledema grade (Frisen grade >3) and cases with progression of papilledema [7]. Although D-dimer is widely used to exclude possible deep-vein thrombosis or pulmonary embolism, it has shown a low sensitivity and specificity as a predictive factor for CVST [8].

Apart from mild-grade disc swelling in the right eye and dyschromatopsia in both eyes, our patient initially had non-significant findings on physical examinations and a non-contrast cranial CT scan. Papilledema, which is usually bilateral unless the other disc is atrophic, raises the possibility of IICP. The bilateral dyschromatopsia of the patient quickly returned to normal after treatment, which may be a result of temporary ischemic injury of the optic nerve. A series of workups for possible hypercoagulable states was performed without significant findings, except for the deficiency of antithrombin activity. However, many factors can affect the accuracy of test results in the acute settings, such as acute inflammation and acute thrombosis [9]. Despite the atypical symptoms or signs and a relatively uncommon patient group in this case, MRI with MRV/MRA of the brain should be considered due to persistent symptoms. Neither observation nor a repeat non-contrast cranial CT scan would be appropriate, owing to new-onset neurological symptoms and relatively low sensitivity of the non-contrast CT scan [2].

## 4. Conclusions

Even though head trauma accounts for only 1–3% of cases of this rare venous thromboembolic disease [4,7], it is important for physicians to raise awareness of the diagnosis of CVST if a patient with definite history of head trauma shows persistent or worsening neurological symptoms despite negative results on serial non-contrast cranial CT scans.

## Figures and Tables

**Figure 1 medicina-58-00140-f001:**
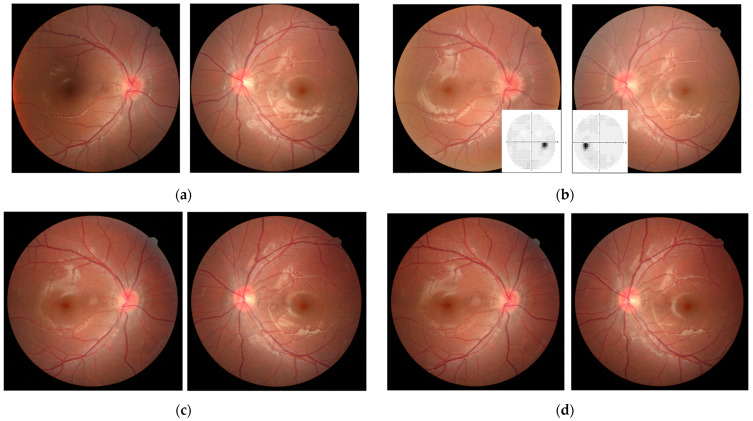
Color fundus photographs of both eyes showed serial changes of the optic disc edema on day 0 (**a**), day 8 (**b**), 3 months (**c**), and 10 months (**d**) post diagnosis of the disease. Automated central 30-2 Humphrey visual field-testing results revealed mild deficits in both eyes ((**b**); insets).

**Figure 2 medicina-58-00140-f002:**
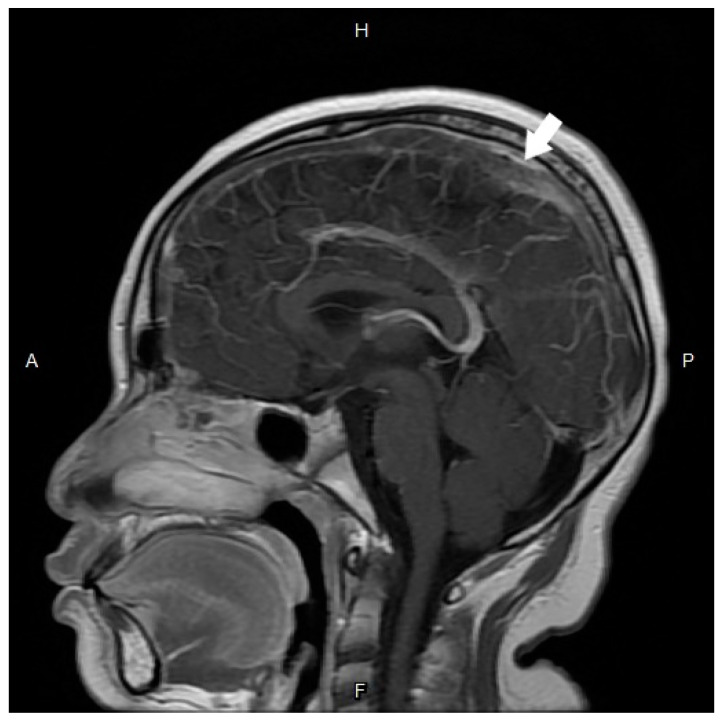
Magnetic resonance imaging with contrast revealed filling defect of the superior sagittal sinus (white arrow) that confirm the diagnosis of cerebral venous sinus thrombosis. A, anterior; P, posterior; H, head; F, foot.

## Data Availability

The datasets used and/or analyzed during the current study are available from the corresponding author on reasonable request.

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
