# Peer review of "Delayed Onset Bilateral Papilledema in a Young Boy’s Eyes after Trauma"

_medicina, 2022, doi:10.3390/medicina58010140_

Round 1
Reviewer 1 Report
- Please define CT in before using the abbreviation to improve clarity for readers.
- Lines 40-41 need clarification.
- The timeline is a bit confusing, especially in line 44.
- Please define "OU" or say "bilaterally"
- Arrows pointing to the thrombosis in the MRI figure may be helpful.
- Was a laboratory workup for a hypercoagulable state performed? Although trauma is the likely etiology in this case it is exceptionally rare. Additionally, with the odd timeframe this other etiologies should be ruled out. I think a workup should be indicated, especially is there is family history. Were there any other pertinents in the history that may make one suspect a different etiology? Perhaps adding a differential diagnosis for CSVT would be beneficial as well.
- Overall, this is an interesting case. To solidify the diagnosis, there should be documentation of ruling out other etiologies of CSVT on the differential diagnosis.
- The manuscript requires a few corrections for the English language to improve clarity.
- The timeline reported can be clarified.
Reviewer 2 Report
Nice case presentation.
Author Response
Response:
Thanks for your kind words and comments.
Reviewer 3 Report
Cerebral venous sinus thrombosis is a rare condition and its clinical presentation is extremely variable. In this case report, the author showed an interesting CVST case with unilateral optic disc edema and vision field deficit in both eyes. A few comments are listed below:
- Line 44: Please specify what kind of medicine.
- Line 73: If available, color fundus photographs of both eyes at 3 months and 10 months should also be presented to make an obvious and sequential comparing.
- Discussion is too short and oversimplified, contents such as standard treatment for CVST or brief summarization of findings from other CVST cases should be added.
- Discussion: Please provide an explanation of dyschromatopsia in this patient.
